# Pan-Cancer Analysis Identifies Tumor Cell Surface Targets for CAR-T Cell Therapies and Antibody Drug Conjugates

**DOI:** 10.3390/cancers14225674

**Published:** 2022-11-18

**Authors:** Xinhui Li, Jian Zhou, Weiwen Zhang, Wenhua You, Jun Wang, Linlin Zhou, Lei Liu, Wei-Wei Chen, Hanjie Li

**Affiliations:** 1State Key Laboratory of Biocontrol, School of Life Sciences, Sun Yat-Sen University, Guangzhou 510275, China; 2Institute of Hepatology, Shenzhen Third People’s Hospital, The Second Affiliated Hospital, School of Medicine, Southern University of Science and Technology, Shenzhen 518055, China; 3Department of Gynaecology and Obstetrics, Shenzhen University General Hospital, Shenzhen 518055, China; 4School of Chemistry and Chemical Engineering, Southeast University, Nanjing 211189, China; 5Department of Immunology, Key Laboratory of Human Functional Genomics of Jiangsu Province, Gusu School, Nanjing Medical University, Nanjing 211166, China; 6CAS Key Laboratory of Quantitative Engineering Biology, Shenzhen Institute of Synthetic Biology, Shenzhen Institute of Advanced Technology, Chinese Academy of Sciences, Shenzhen 518055, China; 7College of Medical Sciences, Qingdao Binhai University, Qingdao 266555, China; 8Shenzhen Key Laboratory of Pathogen and Immunity, National Clinical Research Center for Infectious Disease, State Key Discipline of Infectious Disease, Shenzhen Third People’s Hospital, Second Hospital Affiliated to Southern University of Science and Technology, Shenzhen 518112, China; 9Department of Clinical Oncology, Li Ka Shing Faculty of Medicine, University of Hong Kong, Pokfulam, Hong Kong SAR 999077, China

**Keywords:** pan-cancer analysis, tumor-specific antigens (TSAs), chimeric antigen receptor-T (CAR-T) cell, antibody drug conjugates (ADCs)

## Abstract

**Simple Summary:**

Identification of tumor cell surface targets is vital for chimeric antigen receptor-T cell (CAR-T) therapies and antibody drug conjugates (ADCs). This study utilized the Cancer Genome Atlas (TCGA) database to perform a series of conditional screenings of tumor-specific surface proteins. Accordingly, we found a tumor tissue-specific gene set associated with the survival of cancer patients. Furthermore, these tumor-specific surface proteins can function to render the ability of tumor cells to metastasize. Correlation analysis revealed that these overexpressed membrane proteins were positively correlated, which suggests they maybe potential dual-drug targets. Our findings reveal the significance of tumor cell surface targets in CAR-T- and ADC-related drug development.

**Abstract:**

Tumor cells can be recognized through tumor surface antigens by immune cells and antibodies, which therefore can be used as drug targets for chimeric antigen receptor-T (CAR-T) therapies and antibody drug conjugates (ADCs). In this study, we aimed to identify novel tumor-specific antigens as targets for more effective and safer CAR-T cell therapies and ADCs. Here, we performed differential expression analysis of pan-cancer data obtained from the Cancer Genome Atlas (TCGA), and then performed a series of conditional screenings including Cox regression analysis, Pearson correlation analysis, and risk-score calculation to find tumor-specific cell membrane genes. A tumor tissue-specific and highly expressed gene set containing 3919 genes from 17 cancer types was obtained. Moreover, the prognostic roles of these genes and the functions of these highly expressed membrane proteins were assessed. Notably, 427, 584, 431 and 578 genes were identified as risk factors for LIHC, KIRC, UCEC, and KIRP, respectively. Functional enrichment analysis indicated that these tumor-specific surface proteins might confer tumor cells the ability to invade and metastasize. Furthermore, correlation analysis displayed that most overexpressed membrane proteins were positively correlated to each other. In addition, 371 target membrane protein-coding genes were sifted out by excluding proteins expressed in normal tissues. Apart from the identification of well-validated genes such as GPC3, MSLN and EGFR in the literature, we further confirmed the differential protein expression of 23 proteins: ADD2, DEF6, DOK3, ENO2, FMNL1, MICALL2, PARVG, PSTPIP1, FERMT1, PLEK2, CD109, GNG4, MAPT, OSBPL3, PLXNA1, ROBO1, SLC16A3, SLC26A6, SRGAP2, and TMEM65 in four types of tumors. In summary, our findings reveal novel tumor-specific antigens, which could be potentially used for next-generation CAR-T cell therapies and ADC discovery.

## 1. Introduction

Cancer is the second leading cause of death all over the world with 9.96 million deaths every year [1]. Although tumor cells can be recognized and eliminated by immune surveillance, the immune system may lose efficacy due to the immune suppressive environment induced by tumor cells by altered gene expression, such as mutations and copy number variations [2,3]. The specific proteins encoded by these altered genes expressed on the surface of the tumor cells can shield the tumor cells to evade immune system clearance [4,5]. On the other hand, these tumor specific membrane proteins can be used as drug targets for cancer immunotherapy [6,7], which reactivates the anti-tumor immune response to eradicate tumors.

Nowadays, cancer immunotherapies. such as immune checkpoint blocking therapy (ICB) have been introduced into multiple lines of cancer treatment with great success [8]. Recently, two targeted immune-related therapeutic approaches based on tumor surface antigens (TSAs) have emerged in the cancer immunotherapy research field. The first approach use the antibody drug conjugates (ADCs), a drug class represented by attaching 3–8 molecules of a potent cytotoxic agent to a monoclonal antibody, which targets specific TSAs [9]. The second approach is the chimeric antigen receptor-T (CAR-T) cell therapy [10]. During the T cell engineering process, the expanded T lymphocytes are modified to recognize specific tumor-associated antigens and then transferred back into the cancer patients to eradicate the tumor cells [11]. Current CAR-T therapies mainly focus on CD19 [12], CD20 [13], BCMA [14], MUC1 [15], GD2 [16], CSPG4 [17], HER2 [18], EGFR [19], FAP [20] etc., and have achieved great success in pre-clinical assays or clinical applications. However, there are still many limitations in current TSA-based immunotherapies for tumors, particularly solid tumors [21]. It is mostly due to the lack of specific TSAs (unlike the scenarios in hematologic malignancies, which have specific and well-validated TSAs) [22,23] or heterogenous TSA expression in solid tumors [24,25]. Another major challenge of TSA-based immunotherapies is the “on-target, off-tumor toxicity” effects [26]. Thus, it is quite crucial to identify specific TSAs that are abundantly expressed in tumor cells and less or not expressed in normal tissue cells, to limit the potential toxic and adverse effects.

Here, we exploited the publicly available Cancer Genome Atlas (TCGA) databases to identify specific TSAs in various cancer types. Firstly, we identified highly expressed genes present on the tumor cell surface. Secondly, we performed a survival analysis to select genes that were significantly associated with survival outcomes in cancer patients. In addition, we analyzed the function and correlation of these genes and the prognostic value of the correlated genes with survival rates of cancer patients. To sum up, our work identified specific TSAs that might serve as useful targets for CAR-T cell therapies, ADCs, or co-targeting strategies for the treatment of solid tumors.

## 2. Materials and Methods

### 2.1. Data Source

#### 2.1.1. TCGA

Gene expression of 17 types of tumor samples were collected from TCGA on the University of California Santa Cruz (UCSC) Xena website (https://xenabrowser.net/ (accessed on 1 September 2020)) in fragments per kilobase million (FPKM) (See Table 1 for detailed sample numbers). Human membrane protein information was obtained from the databases: Membranome (https://membranome.org/ (accessed on 1 September 2020)) [27,28] and Uniprot (https://www.uniprot.org/ (accessed on 1 September 2020)) [29]. Human immune cell biomarkers were obtained from the database: CellMarker, and the membrane proteins on the immune cells were excluded [30]. The RNA expression data of 54 human normal tissues were obtained from the GTEx (Genotype–Tissue Expression) project data set (V8 release) [31].

#### 2.1.2. CPTAC (Clinical Proteomic Tumor Analysis Consortium)

The protein expression abundance data of a total of 7 types of cancer tissues were obtained from the CPTAC database. The detailed cancer types and sample numbers are shown in Table 1.

### 2.2. Differential Expression Analysis

#### 2.2.1. TCGA

A rank-sum test was used to analyze the differential expression of membrane protein-coding genes between tumor and normal samples. A threshold of Log2FoldChange (Log2FC) was greater than 1.00 and the adjusted *p* value was less than 0.01.

#### 2.2.2. CPTAC

Taking into account the missing values of protein abundance in the CPTAC data, we first deleted genes whose expression was missing in more than 10% of samples of the tumor (i.e., if there are 100 samples for tumor A, and the expression of gene 1 is missing in more than 10 samples, we deleted the gene), and then filtered the missing values through the K-nearest neighbor method (k = 10). Finally, the relative normalized protein expression abundance profiles of 7 cancers were obtained. Since the protein abundance levels were normalized and log transformed, the difference in the expression abundance was calculated as the abundance in the tumor tissue minus the abundance in the normal tissue via the rank-sum test.

### 2.3. Enrichment Analysis

A hypergeometric test was used to analyze the enrichment relationship between high-expression and high-risk membrane protein-coding genes in the tumor tissues according to the ten cancer hallmarks [32].

### 2.4. Survival Analysis

#### 2.4.1. Log-Rank Test

The tumor patients were divided into a high-expression group and a low-expression group by the mean value of membrane protein expression. A log-rank test was used to compare the survival rates of these two groups of patients using the R package “Survfit” [33].

#### 2.4.2. Multivariate/Univariate Cox Regression

Multivariate Cox regression was used to analyze the impact of membrane protein pairwise combinations on the survival rates of the cancer patients. Hazard ratios (HR) of each membrane protein based on the expression levels of the protein in the sample and the prognostic information of the patient were analyzed through univariate cox regression. Among them, the genes with HR > 1.00 were considered to be poor prognostic factors for the cancer patients.

### 2.5. Risk-Score System Establishment

Each gene score was constructed as the selected gene expression level (*exp*) multiplied by its regression coefficient (*β*) obtained from the univariate Cox regression model. Each patient’s prognostic risk-score was calculated as the sum of two gene scores; the formula is as follows [34,35]:(1)$$Risk\;score=exp_{gene1}*\beta_{gene1}+exp_{gene2}*\beta_{gene2}$$

Based on this formula, the risk-score of each sample was calculated (Appendix A). According to the median risk-score, the patients were divided into a high-risk or low-risk group. The prognostic differences of these two groups were calculated by a log-rank test.

### 2.6. Correlation Analysis

We analyzed the correlation between every two membrane proteins by Pearson correlation. The visualization process was depicted using the R package “corrplot”.

## 3. Results

### 3.1. Identification of Up-Regulated Tumor Cell Membrane Proteins

To analyze the TSAs in tumor tissues, we first identified the membrane protein-coding genes with up-regulated expression levels through a series of pan-cancer screenings. Firstly, the expression profiles of mRNA were obtained from the TCGA database by excluding the data from the cancer types with less than three normal controls (Figure 1A). Secondly, we further selected the tumor cell membrane-coding genes by intersecting the filtered immune cell markers genes and the cell membrane protein-coding genes (Figure 1B). Finally, we examined the potential utility of these membrane genes as drug targets (Figure 1C).

In the first step of the analysis, we found that 3919 membrane proteins were differentially expressed in most tumor samples, compared with their corresponding adjacent normal tissues (Figure 2A and Appendix A). Specifically, we found that the number of up-regulated membrane proteins was larger than that of the down-regulated genes in the 17 cancer types, such as cholangiocarcinoma (CHOL) (up vs. down: 1117:166), stomach adenocarcinoma (STAD) (up vs. down: 252:198), head and neck squamous cell carcinoma (HNSC) (up vs. down: 357:261), kidney renal clear cell carcinoma (KIRC) (up vs. down: 717:427), liver hepatocellular carcinoma (LIHC) (up vs. down: 684:109), lung adenocarcinoma (LUAD) (up vs. down: 478:304) and esophageal carcinoma (ESCA) (up vs. down: 96:78) (Figure 2B and Appendix A). We further illustrated a heatmap to demonstrate the up-regulated membrane proteins in tumor tissues (Figure 2A). Furthermore, a volcano plot shows the differential expression of these target membrane protein-coding genes, which are listed in Table 2.

### 3.2. Most Highly Expressed Membrane Protein-Coding Genes Could Serve as Risk Factors for Cancer Patients

To further explore the impact of these highly expressed membrane proteins of the tumor tissues on the prognosis of the tumor patients, Cox risk regression analysis of these genes was applied to the patients from TCGA (Appendix A). Additionally, we found that most of these membrane proteins were risk factors for tumors. To be specific, 427, 584, 431 and 578 genes were identified as risk factors for LIHC, KIRC, UCEC (uterine corpus endometrial carcinoma), and kidney renal papillary cell carcinoma (KIRP), respectively (Figure 2C). The detailed prognostic values of the membrane proteins have been summarized for further verification in Appendix A. To confirm the reliability of the previous analysis, we searched for whether these target proteins are currently used in TSA-based therapy with solid evidence and found that most membrane proteins we identified were consistent with the published data (Table 3). Furthermore, these tumor cell membrane proteins were either highly expressed in tumors or prognostic risk factors for tumor patients. Furthermore, some of them have been proven to be drug targets in solid tumors, such as GPC3 in liver cancer [36], MSLN in gastric cancer, and EGFR in glioma (Figure 2D).

### 3.3. Function Enrichment Analysis of the Selected Membrane Proteins

To gain deeper insight into the potential functions of these selected membrane proteins, we comprehensively analyzed their functions according to the ten cancer hallmarks, such as invasion and metastasis, etc. [59]. Membrane proteins with higher expression used for functional enrichment analysis are detailed in Table 2. Hypergeometric analysis of these genes demonstrated that they were mainly involved in the invasion and metastasis of the tumor cells (Figure 3).

### 3.4. Paired Membrane Proteins Displayed More Precise Prognostic Value

To further investigate the combinatorial effects of these genes on the survival of the cancer patients, we established a risk scoring system using a formula containing the gene expression levels and the regression coefficients from the univariate Cox regression model (Appendix A). We found that the combined analysis of these genes displayed greater accuracy in predicting the prognostic outcomes of the cancer patients (Appendix A and Figure 4A,B). For example, the prognostic risk stratification power was improved by the following combination of groups, such as *SEZ6* and *ULBP1*, *ULBP2* and *MAFA*, and *PCDHD1* and *MAFA*, compared with the results when the genes were individually analyzed.

### 3.5. The Highly Expressed Cell Surface Proteins of the Tumor Tissues were Highly Correlated

To investigate whether these membrane proteins were correlated, we calculated the association among the membrane proteins in the 17 tumor types through a Pearson’s correlation test. As shown in Figure 5 and Appendix A, the membrane proteins identified previously were significantly positively correlated with most membrane proteins in every tumor type (Figure 5 and Appendix A).

### 3.6. Identification of TSAs That Are Expressed Less in Normal Tissues

To further investigate the potential “on-target, off-tumor toxicity” effect of these proteins, we obtained the expression levels of the 371 target membrane protein-coding genes (See Table 2 for details) from 54 normal human tissues from the GTEx database. The cumulative distribution analysis demonstrated that the TPM expression levels of most genes were logarithmically distributed between −2 and 1.60 (Appendix A). We defined the genes with expression levels greater than or equal to 1.60 as high-expression genes, while genes with expression levels less than 1.60 as low-expression genes. According to this threshold, the genes were divided into two categories (Appendix A). Our results indicated that 184 genes were expressed less in these tissues (Figure 6) and the other 187 genes, which were specifically and highly expressed in some normal tissues (Figure 7). The genes in part one were relatively highly expressed in all brain tissues, while the genes in part two were widely expressed in all tissues. The genes in part three were mainly expressed in the human epidermis, mucous membranes, and glands. The genes in part four were highly expressed in blood cells, lymphocytes, and the spleen (Figure 7).

### 3.7. Validation of Protein Expression of the Selected Genes

To further validate our above findings, we checked the protein expression levels of the obtained genes from the CPTAC tumor protein database (CPTAC, Clinical Proteomic Tumor Analysis Consortium). We identified the expression levels of 23 proteins in four types of tumors (To be specific, KIRC: 8 proteins, LUAD: 3 proteins, LIHC: 11 proteins, UCEC: 1 protein) (Figure 8). It was found that eight proteins (ADD2, DEF6, DOK3, ENO2, FMNL1, MICALL2, PARVG, and PSTPIP1) in KIRC cancer were more highly expressed in tumor tissues (100%), compared with normal tissues. The expression of two proteins (FERMT1, and PLEK2) in LUAD cancer were higher in tumor tissues (66.6%) than in normal tissues. The expression of nine proteins (CD109, GNG4, MAPT, OSBPL3, PLXNA1, ROBO1, SLC16A3, SLC26A6, SRGAP2, and TMEM65) in LIHC cancer were higher in tumor tissues (81.82%) than in normal tissues (Figure 8).

## 4. Discussion

The identification of antigens specifically expressed on the surface of tumor cells is vital for the design of adoptive T cell therapy and ADCs. Our study found that there were various highly expressed membrane proteins in a variety of tumor types (17 cancer types), which were also significantly associated with the survival rate of patients. These findings expand the recent document that identified 200 genes as breast cancer subtype-specific targets by differential expression analysis of RNA-seq data from TCGA [60].

In our discovery, 184 genes were lowly expressed in normal tissues, which further supports the advantage of our strategy in identifying potential targets. Some of them were proved to be successful targets with significant curative effects on some malignant hematological tumors, such as anti-CD19 CAR-T therapy for the treatment of chronic lymphocytic leukemia [61], and anti-CTL019 CAR-T therapy for the treatment of relapsed and refractory B-cell acute lymphoblastic leukemia [62]. In addition, CD66c, CD318, TSPAN8, and CLA were identified as candidate targets for CAR-T therapy in a pancreatic tumor patient-derived xenograft model [63].

Our enrichment analysis identified the relationship of these highly expressed surface proteins and the human immune status, which has a great advantage over the findings from Schreiner et al. Although they could predict the surface antigens of several hematological tumors which may be more applicable across cancer types, they did not exclude the potential adverse effects on immune cells, which may dampen the efficacy of the targets discovery [64]. For example, although the CAR-T therapy targeting CD276 was applied for the treatment of tumors [65], it may cause the death of dendritic cells since CD276 is also expressed in dendritic cells [66]. In addition, our screening methods also excluded CD66c, which was screened out by Schäfer D et al, because of its expression in granulocytes [67]. Furthermore, we excluded the use of the CAR-T target of gliomas: CD70 [68], which can consistently activate T cells and lead to T cell dysfunction [69]. The expression of DLL3 on the tumor tissues of patients with invasive breast cancer can promote the infiltration of immune cells, including plasma cells, CD8 T cells, CD44 memory-activated cells, macrophages, and T regulatory cells [70]. Amir et al., demonstrated that MUC-1 was a target for MUC1-positive cancer cells [58]. Stephen et al. developed an engineered CAR-T cells targeting the HER2+ glioma cells, which also improved disease control in patients with glioma [48]. In sum, our target membrane proteins are ubiquitously lowly expressed in normal tissues. Therefore, it may reduce the immune-associated adverse events during the application of CAR-T or ADCs in cancer treatment [71].

Correlation analysis further demonstrated that the surface proteins with poor prognosis were significantly correlated with each other, which suggests that the effects of the surface proteins on tumor progression may not work independently but coordinate with each other. In general, the paired target membrane proteins could be good potential candidates for dual-target CAR-T therapy and ADCs with fewer side effects [72].

Since the selected surface protein-coding genes were closely associated with each other, we speculated that the selected surface proteins could share a similar transcription pattern. When choosing specific types of surface proteins as drug targets, the off-targets side effects may be prevented by controlling the activation of engineered T cells using integrated multi-input signals [3,73]. Bispecific CAR-T cells targeting PD-L1 and MUC16 have an enhanced killing effect on ovarian cancer cells and significantly prolong the survival time of tumor-bearing mice [74]. CAR-T cells with CEA and MSLN as dual targets can accurately locate the tumor site and have higher toxicity to pancreatic malignancy [75]. Consistently, we found the surface proteins that can be used to design bispecific CAR-T cells or ADCs targeting common solid tumors through big data analysis.

In spite of the limited data in the protein database, we still validated the protein expression levels of some of the selected genes. These results make it more convincing that the identified tumor surface protein genes may be potential targets for CAR-T cell therapies and ADCs. Further integration of proteomic information may boost the discovery of TSAs because the post-modification of proteins such as glycosylation, and lipidation, have also been identified as a source of tumor surface antigens by liquid chromatography–mass spectrometry [76,77].

## 5. Conclusions

In sum, our study revealed some potential tumor-specific surface proteins for the rational design of TSA-based immunotherapies. These findings might pave the way for a comprehensive and efficient approach to construct novel CAR-T cells and ADCs in pre-clinical animal studies and clinical practice by utilizing tumor-specific surface proteins as multi-target binding sites.

## Figures and Tables

**Figure 1 cancers-14-05674-f001:**
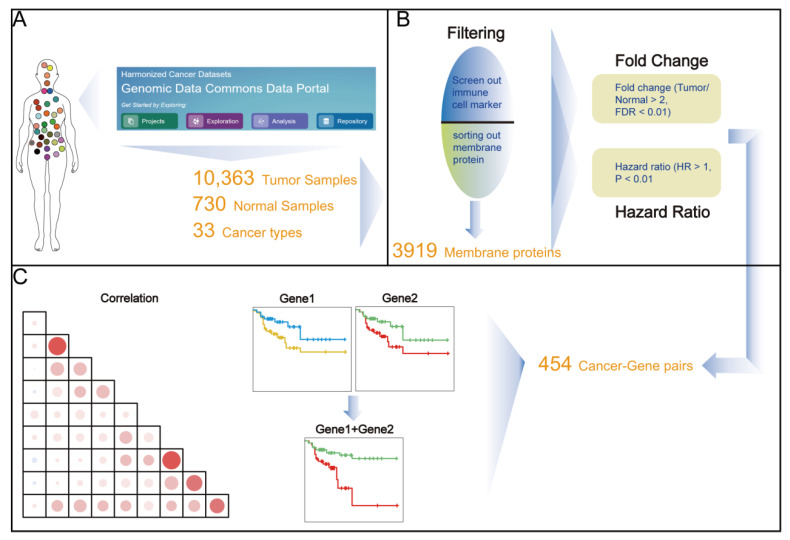
A brief illustration of the core workflow. (**A**) The expression profiles of mRNA were obtained from the TCGA database. (**B**) Thresholds were set to screen the expression profile of membrane proteins (**C**) Visual analysis was used to evaluate the potential functions of the target genes.

**Figure 2 cancers-14-05674-f002:**
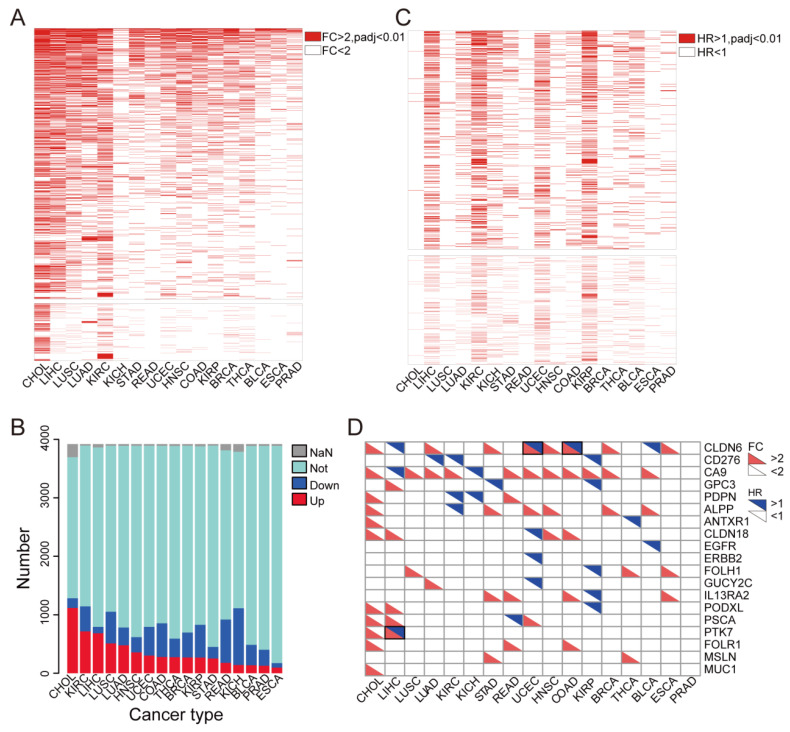
Tumor-specific high-expression and high-risk surface proteins in TCGA tumor tissues. (**A**) Heatmap plot displays the differential expression analysis of the tumor tissue-specific highly expressed membrane proteins. A FC > 2, and the adjusted *p* value less than 0.01 is considered statistically significant. (**B**) Membrane proteins displayed poor prognosis in cancer patients. A HR > 1, and the adjusted *p* value less than 0.01 was considered statistically significant. (**C**) Stacked bar blot showed the categories of 3919 membrane proteins: up-regulation (up), down-regulation (down), no change significance (not), and missing data (NaN). (**D**) The color block diagram shows the FC and HR of the target proteins, all colored squares are statistically significant.

**Figure 3 cancers-14-05674-f003:**
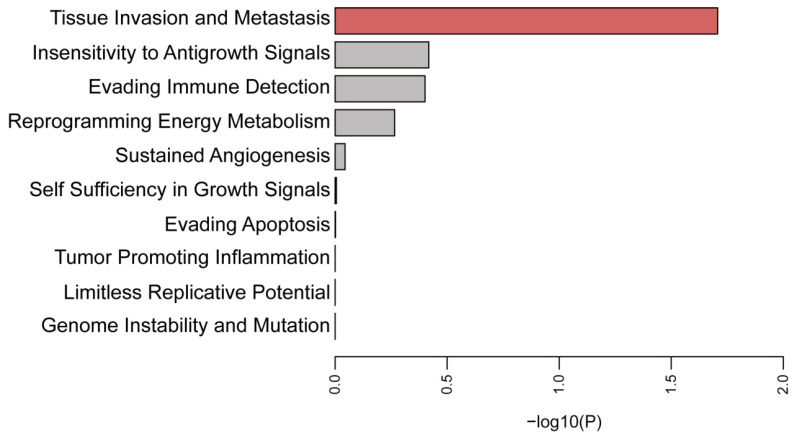
Enrichment analysis of significant high-expression and high-risk genes using the ten cancer hallmarks. The horizontal axis represents the logarithm of the *p* value, and the vertical axis represents the terms of the ten cancer hallmarks. *p* values less than 0.05 were considered statistically significant, indicated by the red color bar.

**Figure 4 cancers-14-05674-f004:**
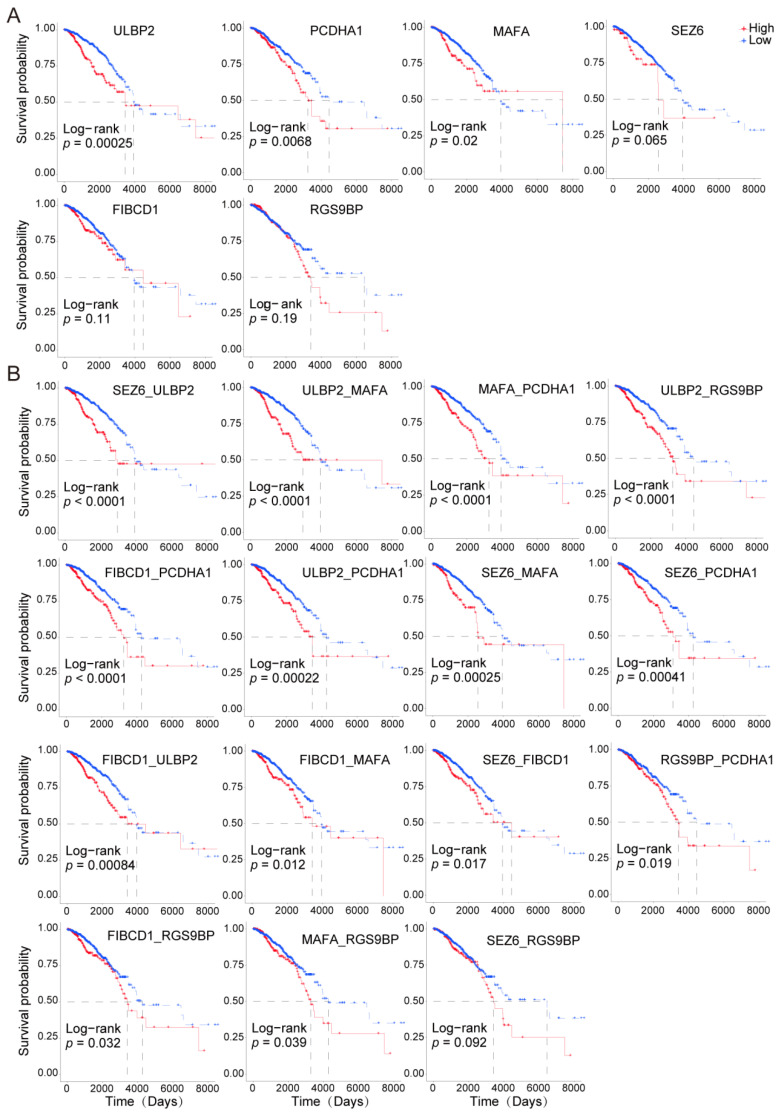
Survival analysis of individual or paired specific high-expression and poor prognosis-associated membrane surface protein-coding genes in BRCA patients. (**A**). KM curves display the prognostic roles of the selected membrane protein-coding genes in cancer patients. (**B**). KM curves display the prognostic roles of the selected membrane protein-coding genes in cancer patients. The red lines indicate the high-expression group. The blue color lines indicate the low-expression group. High-expression and low-expression groups were divided by the mean value of the gene expression. *p* < 0.05 was considered significantly different.

**Figure 5 cancers-14-05674-f005:**
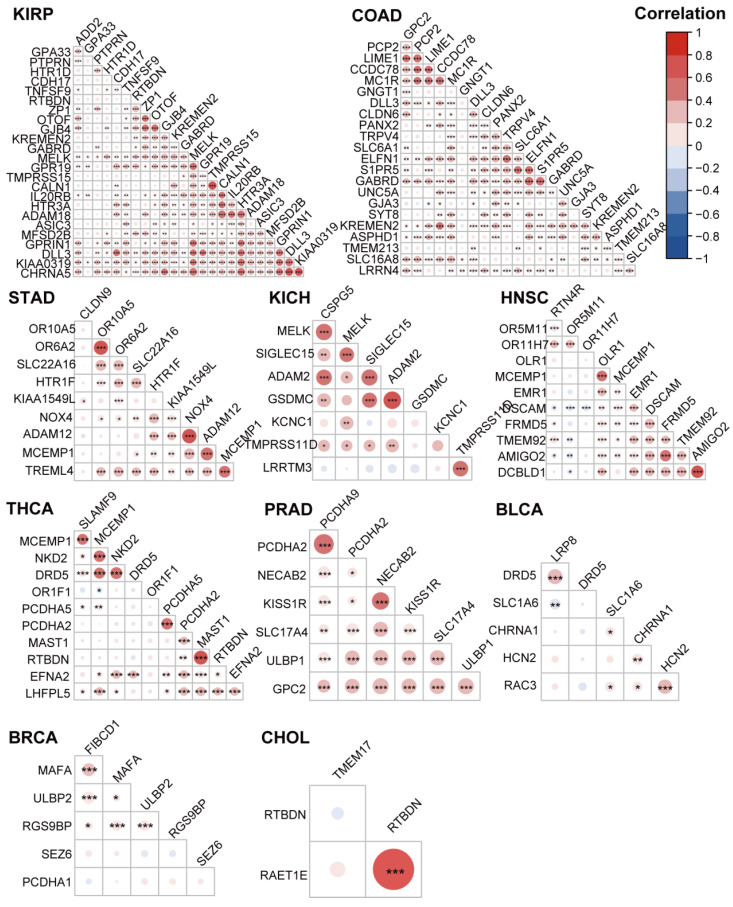
Correlation analysis of the membrane protein-coding genes in different tumor types. The bubble plot shows the Pearson correlation between the high-expression and high-risk membrane protein-coding genes for various cancers from Table 2. The red bubbles represent positive correlations. The blue bubbles represent negative correlations. * means *p* < 0.05, ** means *p* < 0.01, and *** means *p* < 0.001.

**Figure 6 cancers-14-05674-f006:**
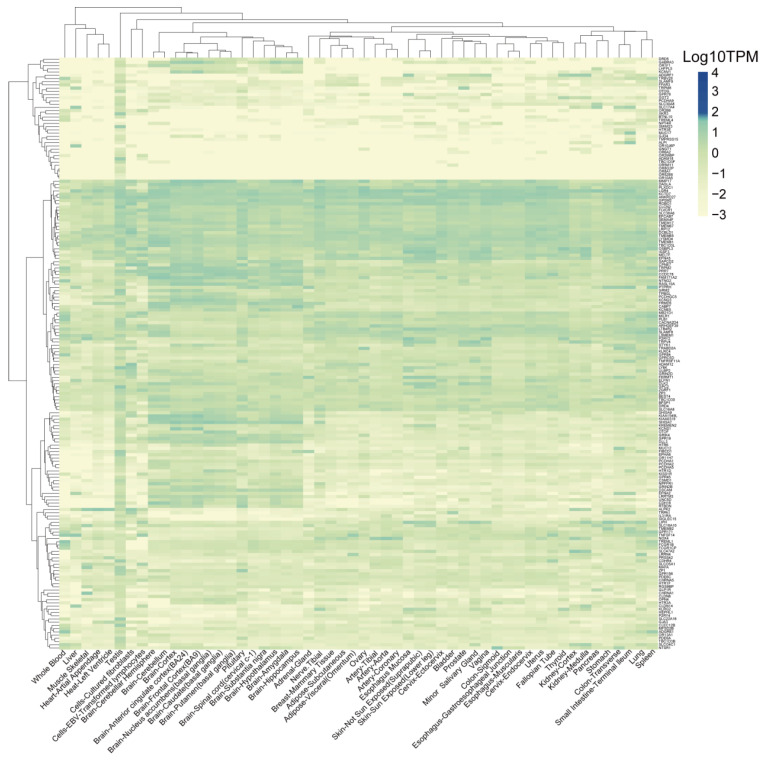
Heatmap displays the expression levels of the low-expression genes in the 54 normal tissues from the GTEx database.

**Figure 7 cancers-14-05674-f007:**
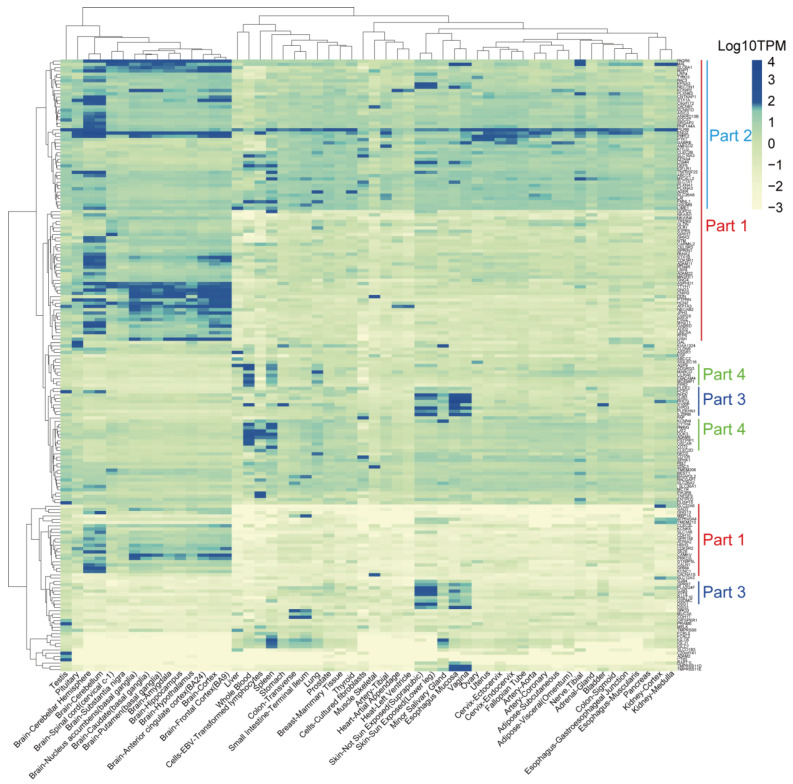
Heatmap displays the expression levels of the highly expressed genes in some tissues of the 54 normal tissues from the GTEx database.

**Figure 8 cancers-14-05674-f008:**
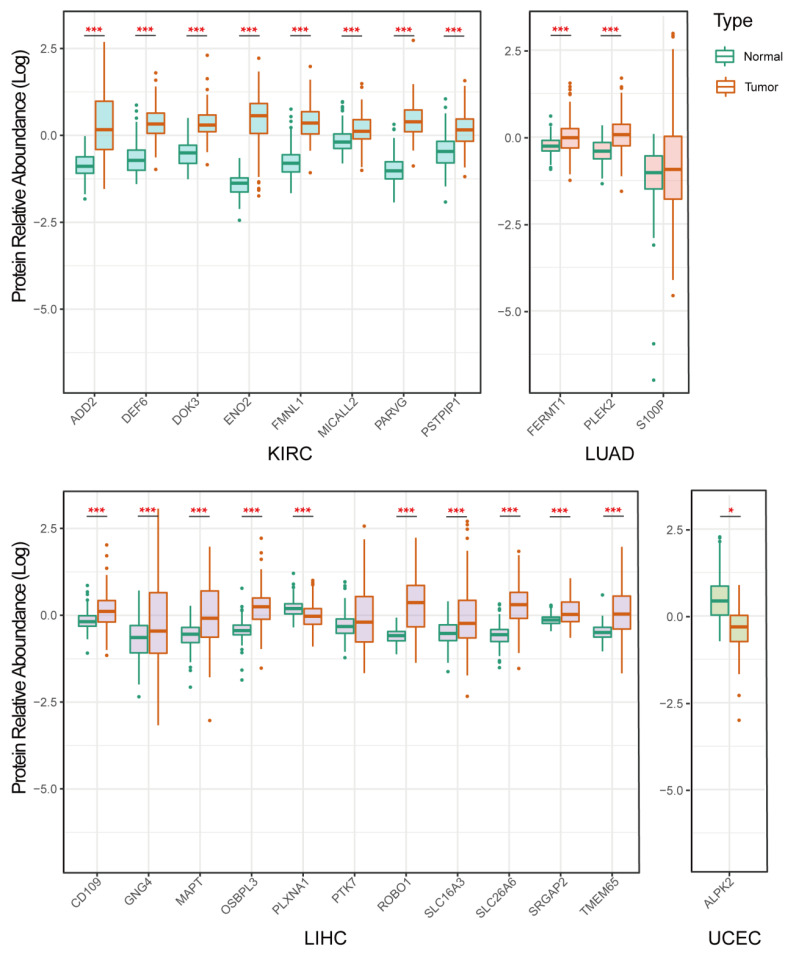
Box plots showing the relative abundance of target membrane proteins in cancer and normal tissues in four tumor types. * means *p* < 0.05, and *** means *p* < 0.001.

**Table 1 cancers-14-05674-t001:** The numbers of tumor and normal samples used in the study.

Cancer Type	Abbreviation	TCGA		CPTAC		
		Tumor Sample	Normal Sample	Tumor Sample	Normal Sample	PDC Study ID
Bladder urothelial carcinoma	BLCA	411	19			
Breast invasive carcinoma	BRCA	1104	113	133	18	PDC000120
Colon adenocarcinoma	COAD	471	41	95	100	PDC000116
Cholangiocarcinoma	CHOL	36	9			
Esophageal carcinoma	ESCA	162	11			
Head and neck squamous cell carcinoma	HNSC	502	44			
Kidney chromophobe	KICH	65	24			
Kidney renal clear cell carcinoma	KIRC	535	72	110	84	PDC000127
Kidney renal papillary cell carcinoma	KIRP	289	32			
Liver hepatocellular carcinoma	LIHC	374	50	165	165	PDC000198
Lung adenocarcinoma	LUAD	526	59	113	102	PDC000153
Lung squamous cell carcinoma	LUSC	501	49	110	104	PDC000224
Prostate adenocarcinoma	PRAD	499	52			
Rectum adenocarcinoma	READ	167	10			
Stomach adenocarcinoma	STAD	375	32			
Thyroid carcinoma	THCA	510	58			
Uterine corpus endometrial carcinoma	UCEC	548	35	100	49	PDC000125

**Table 2 cancers-14-05674-t002:** The surface proteins with high-expression and high-risk in various cancer types.

Cancer	High-Expressed and High-Risk Membrane Proteins in Pan-Cancer	Total
BLCA	CHRNA1, DRD5, HCN2, LRP8, RAC3, SLC1A6	6
BRCA	FIBCD1, MAFA, PCDHA1, RGS9BP, SEZ6, ULBP2	6
CHOL	RAET1E, RTBDN, TMEM17	3
COAD	ASPHD1, CCDC78, CLDN6, DLL3, ELFN1, GABRD, GJA3, GNGT1, GPC2, KREMEN2, LIME1, LRRN4, MC1R, PANX2, PCP2, S1PR5, SLC16A8, SLC6A1, SYT8, TMEM213, TRPV4, UNC5A	22
ESCA	OR2W6P	1
HNSC	AMIGO2, DCBLD1, DSCAM, EMR1, FRMD5, MCEMP1, OLR1, OR11H7, OR5M11, RTN4R, TMEM92	11
KICH	ADAM2, CSPG5, GSDMC, KCNC1, LRRTM3, MELK, SIGLEC15, TMPRSS11D	8
KIRC	ADAM11, ADAM12, ADAM8, ADD2, AGER, AQP9, ARHGEF39, ASGR1, ASIC3, ATP1A3, BEST1, BEST4, BTNL10, C20orf141, C9orf172, CABP7, CACNA2D4, CACNB1, CATSPER1, CCDC78, CD72, CDHR4, CDK5R2, CEACAM4, CELSR3, CLEC12B, CLEC2B, CLEC2D, CNIH2, CNTNAP1, CPNE7, CYTH4, DEF6, DLK2, DOK3, DRD4, EFNA3, EMR1, ENO2, FAP, FCGR1B, FCGR1C, FCRL2, FCRL5, FFAR3, FMNL1, FPR2,GABRE, GAD2, GGT2, GNG13, GPC2, GPR171, GPR19, GPR45, GPR84, GPR97, GPRIN1, GRIK4, GRIN2D, GRM2, GRM4, GSDMB, IGFLR1, IGLC6, IGLC7, IGLJ1, IL20RB, IL31RA, JPH3, KCNIP2, KCNN4, KIAA1324, KLRC4, KREMEN2, LAT, LAT2, LILRA6, LSMEM1, LTB4R, LTB4R2, LY6H, MARCO, MB21D1, MC1R, MCEMP1, MELK, MICALL2, MILR1, MMP17, MUC12, MUC3A, NPFFR1, NPY4R, NTM, NTNG2, OPN4, OR11H7, OR13A1, OR2B6, OSCAR, OTOF, PAQR6, PARVG, PCDHGC5, PDE6C, PIK3R6, PILRB, PLB1, PLEKHN1, PLXDC1, PLXNB3, PRAME, PRKCG, PRR7, PRRT2, PSD2, PSTPIP1, PTPRN, RASL10A, RELT, RLTPR, RTP5, SCNN1D, SIGLEC16, SLAMF8, SLAMF9, SLC16A8, SLCO5A1, SMIM23, SPHK1, STAC3, STX1B, TBC1D3B, TBC1D3F, TBC1D3L, TMPRSS6, TNFRSF25, TNFSF14, TPBGL, TRABD2A, TRBV28, TREML1, TRPM2, TRPM8, TTYH1, ZGRF1, ZP1	148
KIRP	ADAM18, ADD2, ASIC3, CALN1, CDH17, CHRNA5, DLL3, GABRD, GJB4, GPA33, GPR19, GPRIN1, HTR1D, HTR3A, IL20RB, KIAA0319, KREMEN2, MELK, MFSD2B, OTOF, PTPRN, RTBDN, TMPRSS15, TNFSF9, ZP1	25
LIHC	ABCC5, ADAM22, ANKRD13B, ANKRD27, ARHGEF39, ATP8A2, BAIAP2L2, BFSP1, CACNA1S, CD109, CDH10, CDK5R1, CHRNA5, CLCN2, CLEC2L, CNTNAP1, CSMD1, CSPG5, CT83, DAGLA, DDN, DUSP15, EFCAB7, EFNA3, EFNA4, EFNA5EGF, ENTPD2, EPHA6, FAM171A2, FIBCD1, FLVCR1, GABRA3, GDPD2, GJD4, GLDN, GLP1R, GNG4, GNGT1, GPR156, GPRC5D, GPRIN1, GPSM2, GRIN2D, GRM4, GSDMC, IGSF3, KCNE5, KCNQ3, KCTD7, KISS1R, KITLG, LRP12, LRP4, LRP8, LYSMD4, MAGEE1, MAPT, MELK, MEP1A, OR52E6, OR8A1, OR8G3P, OSBPL3, P2RY4, PDE6A, PIK3R6, PLXNA1, PLXNA3, PRR7, PRSS42, PTK7, PTPRN, PVRL1, RACGAP1, RAET1E, RNF144A, ROBO1, SAPCD2, SEMA4F, SHISA7, SLC16A3, SLC22A6, SLC26A2, SLC26A6, SLC30A8, SLC36A1, SLC38A6, SLC47A2, SLC7A1, SOCS7, SRGAP2, STX1A, STXBP5L, TBC1D30, TMEM206, TMEM213, TMEM65, TMEM67, TMEM81, TNFRSF11A, TREM2, TYRO3, XKR3, ZP3	105
LUAD	ABCC2, ALPI, BAIAP2L2, C20orf141, CDH17, CDK5R1, CLDN14, DSG3, ENTPD2, FERMT1, GDPD2, GJA3, GJB2, GJB3, GJB4, GPR37, GPR78, GPR87, GPRIN1, GRIK2, HCN2, HTR1D, IL20RB, KCNV1, KIAA1549L, LGR4, LY6K, LYPD3, MELK, MFI2, OR10J6P, OR1F1, PLEK2, PTPRH, RAET1L, RHCG, RHOV, S100P, SAPCD2, SLCO1B3, STYK1, TMPRSS11E, TRPA1, UNC5D	44
PRAD	GPC2, KISS1R, NECAB2, PCDHA2, PCDHA9, SLC17A4	6
READ	ATP6V0A4	1
STAD	ADAM12, CLDN9, HTR1F, KIAA1549L, MCEMP1, NOX4, OR10A5, OR6A2, SLC22A16, TREML4	10
THCA	DRD5, EFNA2, LHFPL5, MAST1, MCEMP1, NKD2, OR1F1, PCDHA2, PCDHA5, RTBDN, SLAMF9	11
UCEC	ADAM18, ALPK2, ATP1A3, CABP7, CAMKV, CDH18, CDK5R2, CELSR3, CLDN6, CLDN9, DLL3, DSG1, GAL, GNG3, GPR110, GPR158, GPR19, GRIN2B, HEPHL1, HRH3, HTR3A, HTR3E, HTR6, KCNK9, KCNQ3, KCNS1, KIAA1549L, KLRG2, LIPH, MAFA, MAL, MAST1, MEP1A, MUC17, MUC3A, NKAIN1, NKAIN4, NTSR1, OTOG, P2RY2, PLA2G4F, PTPRN, RAC3, SHISA9, SLC12A3, SLC16A10, SLCO4C1	47

**Table 3 cancers-14-05674-t003:** Targets that have been reported for CAR-T therapy in solid tumors.

Protein Name	Cancer Type	Reference
CLDN6	Testicular, ovarian, uterine and lung adenocarcinoma	[37]
CD276	Anaplastic meningioma	[38]
CA9	Metastatic Renal Cell Carcinoma	[39]
GPC3	Hepatocellular Carcinoma	[40,41]
PDPN	Glioblastomas	[42]
ALPP	Colorectal cancer	[43]
ANTXR1	Triple-negative breast cancer	[44]
CLDN18	Gastric cancer	[45]
EGFR	Glioblastoma, Pancreatic carcinoma	[46,47]
HER2	Glioblastoma	[48]
FOLH1	Prostate cancer	[49]
GUCY2C	Colorectal cancer	[50]
IL13RA2	Glioblastoma	[51]
PODXL	Pancreatic carcinoma	[52]
PSCA	Prostate cancer	[53]
PTK7	Lung cancer	[54]
FOLR1	Ovarian Cancer	[55]
MSLN	Gastric cancer, Pancreatic Carcinoma	[56,57]
MUC1	MUC1-positive cancer cells	[58]

## Data Availability

The data sets analyzed during the current study are available from the UCSC Xena website: https://xenabrowser.net/ (accessed on 1 September 2020), Membranome database: https://membran-ome.org/ (accessed on 1 September 2020) and Uniprot database: https://www.uniprot.org/ (accessed on 1 September 2020). Human immune cell biomarkers can be obtained from the CellMarker database: http://biocc.hrbmu. edu.cn/CellMarker/ (accessed on 1 September 2020) and human normal tissues expression data can be collected from the GTEx database: http://gtexportal.org/ (accessed on 1 September 2020).

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
