# Peer review of "Pan-Cancer Analysis Identifies Tumor Cell Surface Targets for CAR-T Cell Therapies and Antibody Drug Conjugates"

_cancers, 2022, doi:10.3390/cancers14225674_

Round 1

Reviewer 1 Report

The present article by Li et al. is very well written and approaches a very relevant topic for targeted immunotherapies. The authors analyze a range of different cancer types using a good number of samples per cancer with its corresponding distal tissue to account for possible toxicities. 

The article is well written. One minor comment is that acronyms should be explained in their first use, both in the abstract and in the main body of the paper.

The findings are well supported by historical data.

Reviewer 2 Report

i found this work extremely well conducted, described and detailed, reaching general relevant scientific conclusions on the possible applicability of novel therapies agaisnt surafce cancer antigens.

Reviewer 3 Report

The article " Pan-cancer analysis identifies tumor cell surface targets for 2 CAR-T cell therapies and antibody drug conjugates" have used TCGA (The Cancer Genome Atlas) databases to identify specific TSAs in various cancer types. They have identified the highly expressed genes on the tumor cell surface and  have performed the survival analysis to select the genes that were significantly associated with the survival outcomes in cancer patients.

The results of this work is very important for selecting targets for CAR-T cell therapies, Antibodies Drug Conjugates, and targeting strategies for the treatment of solid tumors.

The authors must address minor points that can be taken in consideration:

- Introduction section should be improved. 

- Abbreviation should be mentioned in the beginning such as ADCs.

-Table 3 in line 95 was mentioned in the text before table 2. You can arrange that. 

- Discussion section and conclusion can be separated and authors can  highlight more future directions or prospective  for some important genes to be used  for design of TSA-based immunotherapies.
